# The Functions of the Fishermen's Sea Pilgrimage to St Peter and St Paul's Church Fair in the Town of Puck

Tadeusz Palmowski and Lucyna Przybylska *

Institute of Socio-Economic Geography and Spatial Management, Faculty of Social Sciences, University of Gdańsk, 80-309 Gdańsk, Poland
* Correspondence: lucyna.przybylska@ug.edu.pl

**Abstract:** The purpose of this article is to define the function of the Fishermen's Sea Pilgrimage organised in northern Poland. It is the only boat pilgrimage on the Baltic Sea. One of the authors took part in nearly all the pilgrimages in the years 2004–2022. The authors describe the origins and form of sea pilgrimage against contemporary general trends in pilgrimage in Poland. Next, they present the route, participants, and course of this particular event. The participants of the Fishermen's Sea Pilgrimage chose Saints Peter and Paul the Apostles' church in Puck as their destination. The pilgrimage is an illustration of an emerging trend in contemporary pilgrimaging in Poland, characterised by: (1) choosing alternative modes of making the journey, apart from walking; (2) pilgrimage groups not based exclusively on the parishioners' place of residence; (3) the performance of both religious and non-religious functions.

**Keywords:** feretron dance; Puck Bay; sea pilgrimage

## 1. Introduction

The study of religious spaces in Poland dates back to the 1920s and 1930s but was curtailed after World War II due to the communist censorship imposed on research into religious subjects. In post-communist Poland, some geographers started to investigate the pilgrimage tradition in Poland and document the development of pilgrimages to famous sanctuaries and holy places of the Roman Catholic Church (Jackowski 1996, 2003, 2005; Jackowski et al. 1999; Mróz 2014, 2016; Sołjan 2012). Other geographers focused on religious landscapes (Flaga 2011; Klima 2011; Bilska-Wodecka 2012; Przybylska 2014). Polish geographers' studies mostly regard the prevalent denomination, which is Roman Catholicism, professed by 92.6% of Poland's population (Statistical Yearbook of the Republic of Poland 2021). This article refers to Polish Catholics and pilgrimages.

A pilgrimage is a "set of behaviours and rituals in the domain of holiness and transcendence" (Margry 2008a, p. 14). In the religious and spatial sense, the pilgrimage destination is a holy site, as a pilgrimage is a journey "started for religious reasons, to a place considered sacred (*locus sacer*) because of a special agency of a god or deity there, in order to perform specific religious acts of piety and penitence" (Jackowski 2003, p. 120). Maritime pilgrimages are ritual practices that include boat travel over the sea to a location that has historical and/or folkloric meaning, or the pilgrimage place. They emerge "in specific geographical contexts where the population is oriented towards the sea where the basic resources and determinants of the local life, economics, culture and religion are found" (Katić and McDonald 2020, p. 3). We assume that a maritime pilgrimage is a type of pilgrimage. The criterion is the space and the means of transportation ("boat travel over the sea"), as well as the population and the sea interdependence. Harbison (1992), in his book on pilgrimages in Ireland, in the story of the famous voyage of St Brendan the Navigator, uses the terms "sea pilgrimage" and "maritime pilgrimage" interchangeably. In our paper, we chose the notion sea pilgrimage.

Jackowski (2003) distinguishes three basic motives for migrations to cult centres: (1) exclusively religious, (2) religious-cognitive, and (3) non-religious ones. He believes that in the first two cases, we deal with religious tourism, while in the third case—due to the purely secular motivation to travel—with cultural tourism. Establishing why a person sets out on a pilgrimage along a specific trail is possible through an interview or a questionnaire. Motivational dualism, i.e., combining religious and cognitive (tourist) motivations, commonly occurred in pilgrimages on Polish territory in the past and it still does today. An expert on Polish pilgrimages distinguishes their additional functions, such as expressing patriotic sentiments and community building in the religious, social, as well as national dimensions (Jackowski 2003).

The purpose of this article is to define the function of the Fishermen's Sea Pilgrimage—an event organised in northern Poland. It is the only boat pilgrimage on the Baltic Sea and, at the same time, a distinct form of pilgrimage among Christians living around the world. For instance, in Chile, on St Peter's Day, fishermen sail to Valparaiso, fishermen from Rameswaram in India travel to the island of Katchatheevu near the coast of Sri Lanka, and in China, fishermen go on a pilgrimage by sea to the She Shan Marian sanctuary near Shanghai (Marciniak 2010). In Europe, sea pilgrimages take place in Spanish seaside destinations, such as Huelva, Santurce, Rincón de la Victoria, Marbella, El Palo, and Pedregalejo, as well as in Tenerife, Majorca, and Minorca (Huras and Necel 2017). There are also maritime pilgrimage traditions in Croatia, Monte Negro, Portugal, and Ireland (Katić and McDonald 2020).

The Fishermen's Sea Pilgrimage in Poland has been documented in photo albums (Huras and Necel 2017, 2021; Kownacka 2014), published in the form of reports, reportages, and discussions (Celarek 1981; Czerny 2000; Miluszewska 2002; Marciniak 2008, 2010; Lemak 2003). It was also the object of research conducted by Marcin Turzyński (2005) as a part of his MA thesis on Apostolatus Maris (the Apostleship of the Sea), written under the supervision of one of the authors of this article. To the best of their knowledge, literature regarding the Fishermen's Sea Pilgrimage has been published only in Polish and mostly in journals by regional publishers, except for works in Polish that appeared in the interdisciplinary journal *Peregrinus Cracoviensis* (Czerny 2000; Turzyński 2006).

In general, the article can be described as a review paper. When working on it, the authors used participant observation as a supplementary method. One of them (Tadeusz Palmowski) took part in nearly all the events in question held from 2004 to 2022. The photographic documentation compiled over that time was used by Aleksandra Tarkowska (2016) in her work entitled *Secrets of Hel Peninsula*, as well as by the other co-author, in her work devoted to the sacralisation of public spaces in Poland (Przybylska 2014). Sharing the experience of the pilgrimage and the conversations held on board the fishing boat JAS 57 Magdalena[1] with Edward Muża—its owner, skipper, and a fisherman—as well as his family, enabled the authors to gain in-depth knowledge about this exceptional pilgrimage and draw conclusions regarding its functions.

In the first section of the article, the authors describe the origins and form of sea pilgrimage against contemporary general trends in pilgrimage in Poland. Next, they present the route, participants, and course of this particular event. The article closes with conclusions.

## 2. The Origins and Form of the Fishermen's Sea Pilgrimage against Contemporary Trends in Pilgrimaging in Poland

Pilgrimaging in Poland is a centuries-old tradition, starting from the religious journeys made in the late 10th century to the grave of the first martyr on Polish lands (St Adalbert's tomb in Gniezno), as well as to the graves of other people who died surrounded by the atmosphere of holiness. Later, in the 12th century, pilgrims started to travel to the Holy Rood church in the mountains, where a relic alleged to be part of the True Cross was held (Jackowski 2003). In contemporary times, and beginning in the mid-17th century, the most important sanctuary for Polish Catholics is the National Shrine of Our Lady in Częstochowa.

During the period of the Partitions (1795–1918), shrines and pilgrimages began to be related to a Polish national identity. The shrine in Czestochowa "gained a leading role as 'national sanctuary': a place where a 'real Queen' (as opposed to partitioning usurpers) 'resided' and could unite Poles arriving at the shrine from the three partitioned regions" (Niedźwiedź 2015, pp. 74–75).

According to Jackowski (2003, p. 232), "after 1945, pilgrimages in Poland went through different phases: from their growth right after the war, through a clear regression in the 1950s and 1960s, to a rapid development that started in the 1970s". It should be remembered that, although pilgrimages were organised in Poland in the socialist era, the participation of state officials or soldiers in those events, like in other religious practices, had to be kept secret. Moreover, pilgrimage organisers and church builders had to deal with obstacles created by state authorities because administrative ordinances were issued to "hinder or even disable the organization of pilgrimages, especially walking pilgrimages, to sacred places" (Mróz 2021, p. 17). Thus, apart from having a religious dimension, participation in a pilgrimage was also an expression of rebellion against the authorities' official, anticlerical, and anti-liberty propaganda.

The difficulties that had to be overcome by the organisers of the Fishermen's Sea Pilgrimage were described by Czerny (2000). After its first successful inauguration in June 1981, Polish fishermen were not permitted to organise the next sea pilgrimage, due to the martial law introduced on 13 December 1981 in Poland. Travelling was restricted. While the first pilgrimage included 42 boats, in 1982, only 26 sailed to Puck (Czerny 2000). They were welcomed by the parishioners and the priest, but also by army men and militiamen, who took down the numbers of the boats; their owners were later called for questioning, but ultimately had to pay small fines only. Furthermore, the soldiers who did not report as witnesses were punished. The pilgrims defended themselves saying that they had not violated the law forbidding nautical tourism but participated in a religious cult.

After the collapse of communism in Poland in 1989, social, political, and economic transformations were followed by changes in religious tourism. Gained in the 1980s, religious freedom, as well as the growing prosperity and mobility of Polish society in subsequent decades, had a significant impact on pilgrimaging as a result of joining the European Union in 2004. The followers of the Roman Catholic Church in Poland go on pilgrimages individually or participate in religious journeys organised by parishes or other social groups, such as physicians, coal miners, youth, etc. Let us quote the example of the most frequented sanctuary in Poland—the National Shrine of Our Lady in Częstochowa. Some pilgrimages have a long tradition, e.g., the Warsaw Pilgrimage, organised since 1711 (Niedźwiedź 2015; Mróz 2021), but many large events started in the 1980s and 1990s. For instance, in 2022, the Parliamentarians' Pilgrimage took place for the 33rd time, the Polish Veterinarians' Pilgrimage, for the 10th time, or the John Paul II Pilgrimage, organised by schools bearing his name, for the 22nd time. The Fishermen's Sea Pilgrimage fits this model. It started in 1981 and gathers a homogeneous group of sea fishermen. The main organiser of the event was Aleksander Celarek, an engineer and inhabitant of Chałupy (a village on Hel Peninsula), a sailor, shipwright, and sailmaker (Celarek 1981; Czerny 2000). During his trip to South America, he met a school friend who was a priest and a missionary in Brazil. The friend told him about a tradition of Brazilian fishermen who decorate their boats on the Day of St Peter the Fisher and sail on the Amazon River to venerate their patron saint. That story inspired Celarek to organise the Fishermen's Sea Pilgrimage in Poland.

In the last three decades, the number of Roman Catholic sanctuaries in Poland has grown significantly. While in the 1990s there were over 500 of them (Jackowski 1996), in 2016, the number increased to 820 (Mróz 2016). Both formerly and currently, the majority of Catholic sanctuaries are Marian shrines. In 2014, the major (international, national, and supraregional) sanctuaries in Poland were visited by over 12 million people (Mróz 2014). The church in Puck, where fishermen sail on a pilgrimage every year in June, does not have the status of a sanctuary, according to the Code of Canon Law (Biskupski 1984, canon 1230). It is a parish church dating back to the 14th century, which has the additional function of

the deanery seat. Saints Peter and Paul the Apostles' church is a sacred place, because, by the Code of Canon Law (Biskupski 1984, canon 1205), sacred places include churches, chapels, sanctuaries, cemeteries, and altars, which "by consecration or blessing performed in accordance with liturgical regulations are dedicated to the cult of God or used as the followers' burial ground". Perhaps in the future, due to the Fishermen's Sea Pilgrimage, the church in Puck will be granted the status of a sanctuary, after receiving the "approval of the local ordinary", which is required to gain canonical recognition. It is worth noting that over half of the 820 sanctuaries in Poland do not have a clear legal status in this respect (Mróz 2016).

As it was mentioned at the beginning of this section regarding the long tradition of pilgrimages in Poland, the Fishermen's Sea Pilgrimage also dates back to the old times. Already in the 13th century, boats from Hel Peninsula sailed to visit the annual church fair in Puck devoted to the parish's patron saints. It is worth mentioning the geographical conditions of this custom. Initially, it was possible to reach Puck only by fishing boat. With time, Hel Peninsula was enlarged by sand sediments brought by the sea current and turned from the original group of isles into a sandbar peninsula (Augustowski 1977). It must be stressed that before World War II, fishermen's boat pilgrimages set out from the ports and marinas of Hel Peninsula and Rewa to Swarzewo. Our Lady of Swarzewo, alternatively called the Queen of the Polish Sea and the Protector of Kashubian Fishermen, has been the object of the cult since the 16th century (Kownacka 2014; Mazurek 2021). The dilemma of whether to sail to Swarzewo or Puck was solved by the parish priest from Kuźnica, Gerard Markowski, who pointed to Puck, because Swarzewo is the destination of the traditional, overland Maszopska Pilgrimage, organised every year from Hel Peninsula (Miluszewska 2002). The idea of a sea pilgrimage was approved by fishermen, who declared their readiness to carry the willing pilgrims on their boats (Marciniak 2010).

Modern pilgrimages in Poland reflect more extensive cultural changes, including European societies' love of individualism and tourism. The new cultural and religious tourism trails created in the 21st century (Ways of St James and Papal Trails) are examples of, if not the individualisation of religious practices, then at least the popularisation of a new form of pilgrimage in Poland—participating individually or in a small group. What is more, one may have an impression that the journey itself is more important than the destination—the sacred place. Similar conclusions were drawn by Margry (2008b) when he discussed the status of the Way of St James in contemporary Europe. In Poland, part of the old Christian tradition of travelling from home to the grave of St James the Apostle in Spanish Santiago de Compostela, over 5500 km of roads have been marked out since 2005. As a result, "currently, the Way of St James in Poland is the longest pilgrimage, cultural and thematic trail in Poland" (Mróz 2014, p. 133). Papal Trails, mostly hiking and kayaking ones, propagate John Paul II's teachings and style of recreation (Własiuk 2010, p. 328).

The kayaking trails mentioned above are not the only innovation available for the contemporary pilgrim in terms of transport. It must be stressed that new forms of pilgrimage appear next to the traditional walking pilgrimages and popular pilgrimages by coach or car. A geographer from Cracow, Mróz (2014), distinguished cycling, motorcycling, caravanning, skiing, horse riding, Nordic walking, individual and relay running, roller-blading and roller skiing, and even paragliding pilgrimages. He did not mention the boating form, which occurs in Poland only at two places in the north of the country: on Żarnowieckie Lake and on the waters of the southern Baltic Sea, in Puck Bay. It must be emphasised that the Fishermen's Sea Pilgrimage does not represent a networkisation either (Przybylska 2014). In this case, good examples are the Papal Trails or Way of St James. The Fishermen's Sea Pilgrimage is unique; it refers to one space and one time and has no equivalent in another sub-sea basin in the country. It does not have network features characteristic for another modern trend regarding pilgrimage and religious events in Poland.

### 3. The Fishermen's Sea Pilgrimage—Time, Space, Participants, and Their Activities

On 29th June, the Catholic Church celebrates St Peter and St Paul's Day. Going on a pilgrimage to St Peter and St Paul's church fair in Puck has traditionally taken place on a Saturday—a free day for fishermen and visitors. The only exception was the year 2001, when the pilgrimage was organised on a Friday. Every year, since the first pilgrimage in 1981, pilgrims have set out to the church in Puck across the Small Sea, as Puck Bay is sometimes called. The boats sail from all the ports and marinas of Hel Peninsula and Puck Bay (Figure 1). The longest distance has to be sailed by the fishermen from Władysławowo, who go around the 35-km-long peninsula. Every year, on the day of the pilgrimage, early in the morning, one of the co-authors arrived at the port in Jastarnia, where fishermen were preparing for the trip. At 9 o'clock, a fleet of fishing boats sailed in tight formation from Jastarnia to the waters of the Bay. The safety of the pilgrims was guarded by the Navy, Border Guards, Maritime Office, and Polish Ship Salvaging vessels. Due to the bathymetric and navigational conditions, the pilgrims first sail towards Rewa, then Swarzewo, where they are joined by other larger and smaller vessels, some of them under sail. Those are often *pomerankas*, constructed by Celarek (2000)[2].

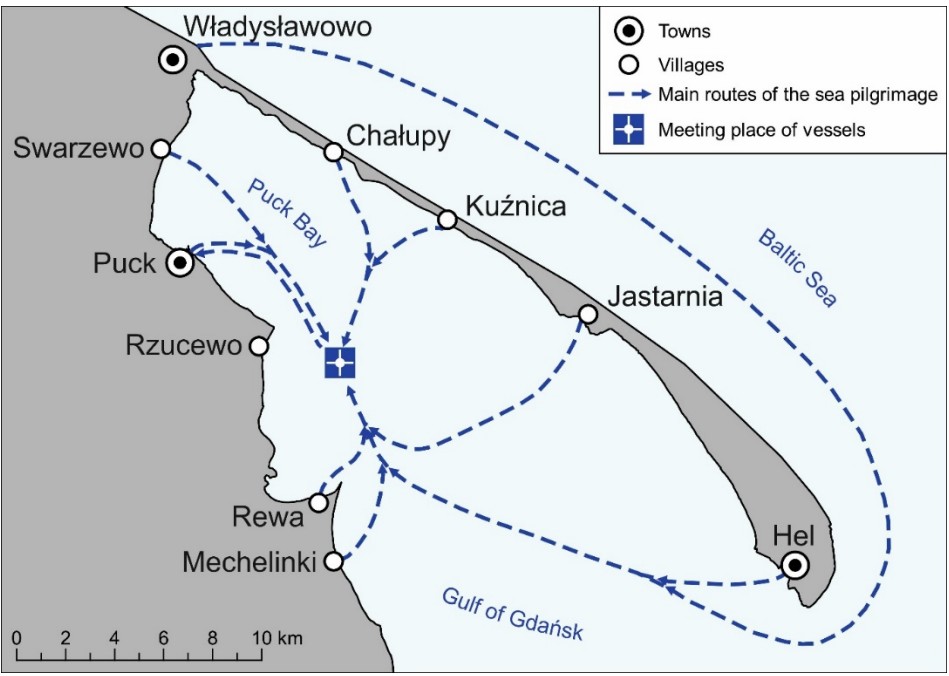

**Figure 1.** The route of the annual Fishermen's Sea Pilgrimage. Source: authors' elaboration.

The number of vessels participating in the pilgrimage in individual years varied depending on the political and weather conditions (Turzyński 2006), and recently also due to the pandemic. The smallest numbers were recorded in the years of martial law (1982–1983) and during the COVID-19 pandemic (2020). Typical vessels taking part in the sea pilgrimage are presented in Figure 2. The largest numbers of fishing boats participating in the pilgrimages were recorded in 1991–1992, 1997–1998, and 2010, exceeding 50. In other years, the number ranged from 30 to 50. The fishing boat *Magdalena* (Figure 3 on the right) usually carried over 40 people. Slightly fewer boats took part in the pilgrimages organised in the second decade of the 21st century (Huras and Necel 2021). It is difficult to establish the number of all pilgrims in individual years, but it can be assumed that in the years in which over 50 vessels took part, the number of participants might have reached 1200 people.

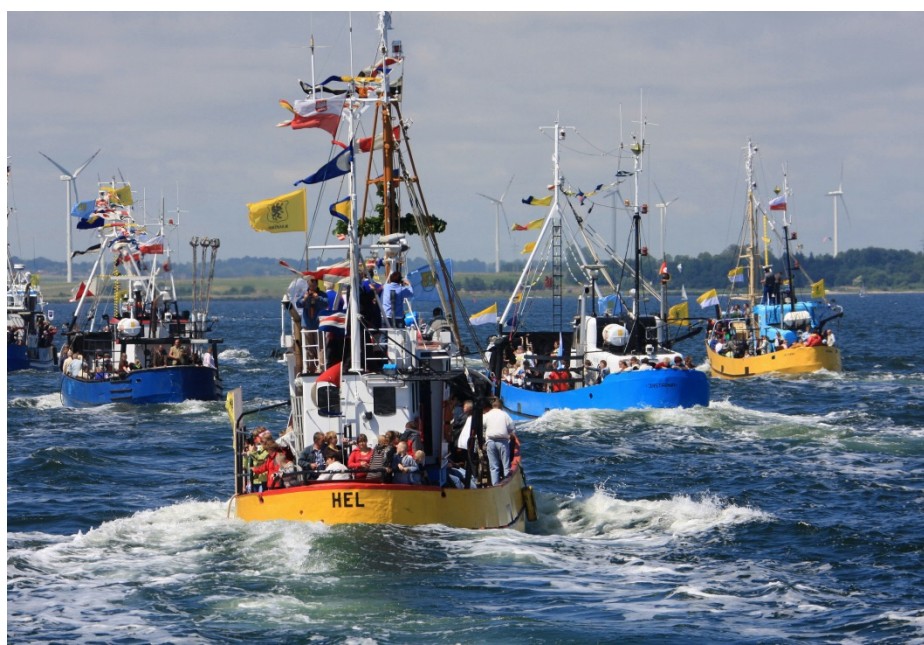

**Figure 2.** Typical fishing boats participating in the sea pilgrimage. Author: (Tadeusz Palmowski 2009).

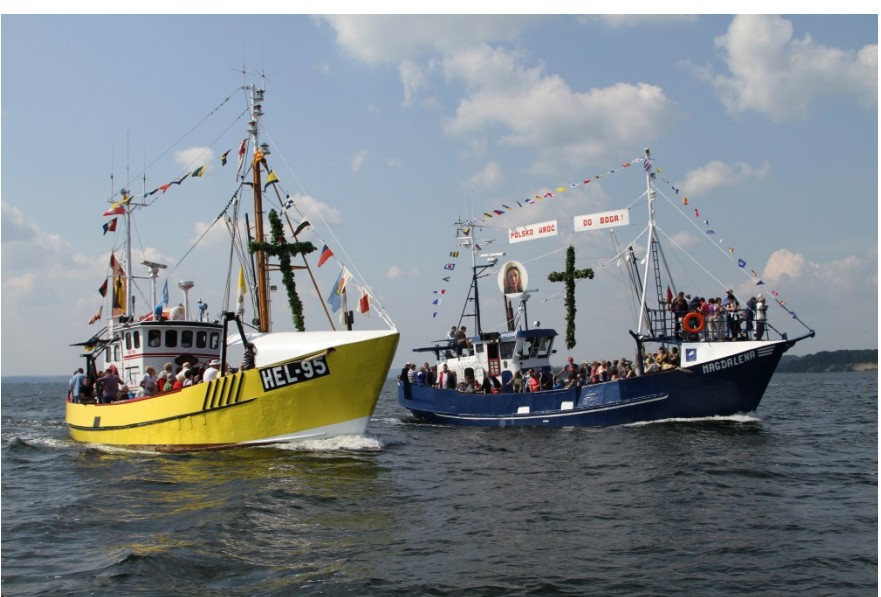

**Figure 3.** *Hel-95* and *Magdalena* fishing boats. Author: (Tadeusz Palmowski 2013).

The founder of the pilgrimage (Celarek 1981) describes the first pilgrimage and, at the same time, the new tradition of having a religious service on the water before the pilgrimage reaches Puck. He remembers the morning of 28 June 1981, when 42 fishing vessels from Chałupy, Kuźnica, and Jastarnia, festively decorated with white and red national flags and white and yellow church flags and bouquets of flowers, with over 300 pilgrims on board, sailed towards Puck. They were accompanied by a crowd of people, standing on the shore and waving. The fleet was led by a boat with a large birch cross placed upright on board (Celarek 1981). To avoid arriving in Puck too early, all the boats were set adrift at about 11 o'clock, mooring around the leading boat. The pilgrims spent that time singing religious songs. After an hour, with delicate gusts of wind, the whole fleet sailed towards Puck. That event started a new tradition—since 1985, boats sailing from all ports and marinas have met on the route perpendicular to Rzucewo. There, large fishing boats moor side to side, and then they are surrounded by smaller boats, forming a "floating

island" (Figure 4). After that, boats from Puck arrive, bringing the bishop and honorary guests. After welcoming the pilgrims, the bishop performs a short service, during which he emphasises the deep faith and the permanence of Kashubian fishermen's hard work on the sea every day. In the last two decades, the service on the water started at about 11.45. After that, the impressive, colourful armada sailed to the port in Puck (Figure 5), where large fishing boats moored first, followed by the remaining vessels. After the pilgrims were welcomed, at about 12.30–13.00, very near the port, the Indulgence Holy Mass was held. It is estimated that, in 2004, the celebrations and the pilgrimage were attended by 3000–4000 people (Turzyński 2006).

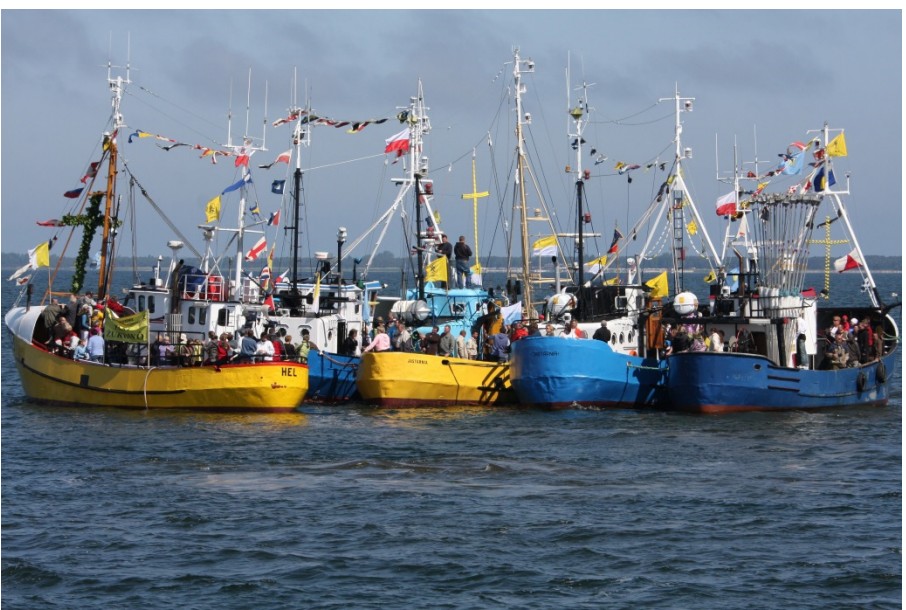

**Figure 4.** Fishing boats meeting on Puck Bay to participate in a Holy Mass together. Author: (Tadeusz Palmowski 2009).

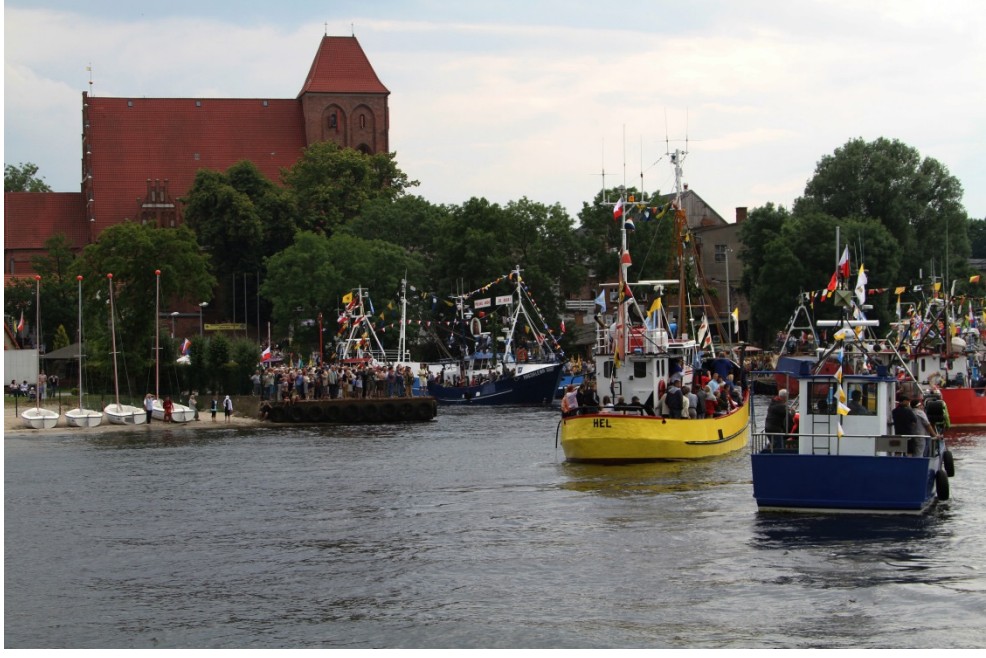

**Figure 5.** Fishing boats docking at the port in Puck. Author: (Tadeusz Palmowski 2013).

We should stress the atmosphere of concentration and prayer during the sea journey to Puck. Using the loudspeakers fitted on the vessels, all the pilgrims join in prayer together. They also sing religious songs to the sound of the guitar, accordion, or sometimes even an orchestra present on board. When it is time for saying the Rosary, we can see how many pilgrims are holding the beads, moving them with their fingers. During breaks in the "religious program", the co-author of this article was often offered coffee and homemade cake by Barbara, Edward Muża's wife. Moreover, we should note the fishermen's festive outfits and the decoration of their boats (Figures 3 and 4), the flag gala, i.e., the flags of the international sea code, national, papal, and Kashubian flags, as well as wooden crosses, often adorned with oak leaves (Figure 3). We can see holy pictures and religious slogans, such as "I want to be closer to You", "Poland, return to God" (Figure 3), or "Mother of the Sea, bless fishermen and us". There are also banners referring to contemporary events important for the fishermen, such as imposing certain fishing restrictions to protect porpoises, or introducing general fishing limits.

In Puck, after the pilgrims leave the boats to be welcomed and later, when the inhabitants bid farewell to them, we may notice a unique element of the pilgrimage tradition in the ethnographic region of Kashubia (Kashubia 2022), occurring not only in the northern fragment of Puck County, shown on the map in Figure 1, but also in some other counties in Pomeranian Voivodeship (Mordawski 2018). This unusual ritual is the *feretron* dance, i.e., the bowing of portable holy images carried by the pilgrims (Weiher-Sitkiewicz 2021). Figure 6 depicts the bow of a *feretron* with the image of the Virgin Mary. Other *feretrons* presented the images of Saints Peter and Paul the Apostles, as well as Christ holding out his hand to save Peter from drowning. In this particular type of dance, found only in Kashubia, the persons who carry the *feretron* tilt and rotate it dynamically, moving it along a cross or circle path, and finally making a deep bow forward. This "dance" raises extreme astonishment and great interest among all the pilgrims and visitors, particularly those who are watching this unique event for the first time.

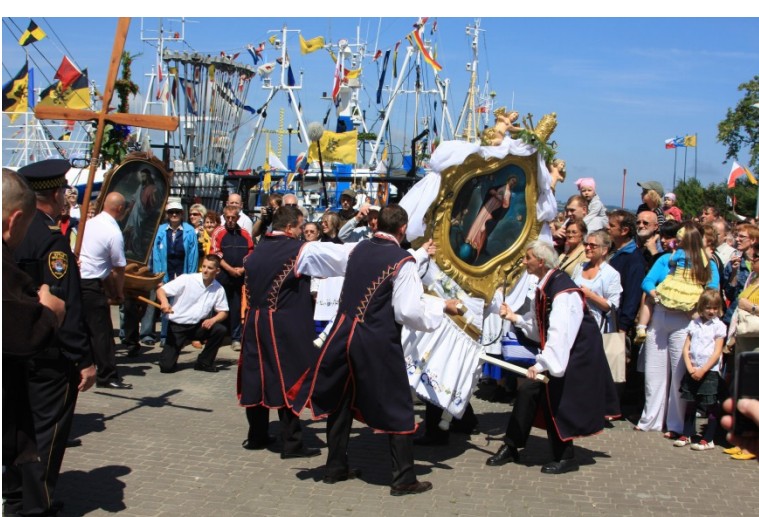

**Figure 6.** The *feretron* dance in Puck. Author: (Tadeusz Palmowski 2009).

After the welcoming *feretron* dance, delegations with *feretrons* and church banners, the banners of the Kashubian–Pomeranian Association divisions from the whole Pomerania region, the Polish Sea Fishermen's Association banner, as well as other flags, position themselves on both sides of the field altar, placed between the church and the port. On the right side, facing the altar, there are the choir and the orchestra, whose members originate not only from Puck, but also from other parts of Kashubia. During a Solemn Mass, they thank God for the goods they have received and pray for the prosperity of all seamen. When it is time to bring offerings, fishermen in festive traditional Kashubian clothes bring fish which have been caught by them in the Baltic sea and smoked. It should

be stressed that some parts of the Holy Mass are celebrated and some songs are sung in the Kashubian language, which has the status of a regional language in Poland, despite the fact that "the Kashubs themselves as a community do not have a recognised status" (Mazurek 2021, p. 12). The Kashubian identity is based on language, origin, territory, and religion[3]. In the national census of 2011, 108,140 people declared that they spoke the Kashubian language at home. In the years 2020–2021, the Kashubian language was taught in 319 primary schools (Statistical Yearbook of the Republic of Poland 2021, p. 347). For a Polish sociologist, Monika Mazurek (2021, p. 8), "the Kashubian language is the greatest symbolic value for Kashubian identity and a ritual trigger of ethnic identity". Moreover, she emphasises the process of ethnicisation of religion, which means, in regard to Kashubs, the use of the Kashubian language or other elements of Kashubian culture in the religious rites.

After the Holy Mass, all pilgrims named Peter and Paul gather near the altar, where photographs to remember the occasion are taken. Next, at the same place, but in a more relaxed atmosphere, pilgrims can listen to concerts given by Kashubian and other choirs, bands, and solo singers, and near the port and in the town streets, they can visit a multitude of stands and stalls selling pieces of traditional Kashubian art. After that, always at 5 p.m., on the port embankment filled with visitors, the fishing boats and pilgrims are blessed. On the way back, with fewer guests on board (some pilgrims return home by land directly from Puck), individual boats return to their mother ports, this time not in tight formation. In Jastarnia, they moor at around 7 p.m.

A survey conducted on 27 June 2004 by Marcin Turzyński (2005) on five randomly chosen fishing boats setting out from Jastarnia (74 people) made it possible to create a profile of the Fishermen's Sea Pilgrimage participants. The pilgrims participating in the survey included inhabitants of 26 destinations, who were members of 46 parish communities. Most pilgrims were the inhabitants of Jastarnia, and most participants in the Holy Mass held in Puck were residents of this town (about 75%). Apart from the inhabitants of the nearby agglomeration, Gdańsk, large groups of pilgrims arrived from all over Poland, e.g., from Białystok, Warsaw, Bydgoszcz, Olsztyn, or Włocławek. Most pilgrims were women (65%). Nearly 50% were aged 18–50, 38% were over 50, and only 10% were under 18. Generally speaking, the participants of the pilgrimage were middle-aged and elderly, and often belonged to organised groups, travelling even from distant destinations. The majority of them had completed secondary (46%) or primary (23%) education. University education (18%) was declared mostly by the participants arriving from large cities, such as Gdańsk, Gdynia, or Warsaw. The largest group included working people (36%), followed by students and retired persons (26–27% each). The vessels included in the survey carried about 30–40 people on board. Fishermen often invite various organised groups, not necessarily associated with the sea—in this survey, as many as 55% declared the lack of such associations. 14% of the remaining pilgrims were dock workers, sailors, or retired workers of the maritime sector, 11% were fishermen, and 14% were their relatives and housemates working on the sea. This leads to a rough estimate that people of the sea made up less than half of the pilgrims.

In 2004, the majority of the pilgrimage participants were believers (57%) and people who were deeply devout (36%). Nearly 84% of them were practising Catholics. Non-practising people or non-believers were not found among the participants (Turzyński 2005). The Fishermen's Sea Pilgrimage attracts even those who do not belong to any church organisation (65%). The group included young people arriving from far away in their quest for God, to build and deepen their faith, and find its witness in the religious zeal of the Kashubs. The spiritual need encouraged over 55% of the pilgrims to take part in the pilgrimage, and over 33% arrived there out of curiosity. Some (e.g., the Missionary Sisters of St Elisabeth) treated this pilgrimage as a kind of retreat, while others joined because of their attachment and affection for the sea and the people associated with it. The religious reason to go on a pilgrimage was expressed by nearly 88% of the respondents. There were also people guided by the cognitive motive (24%), and only 4% treated the

pilgrimage as an occasion to rest and experience customs they did not know (Turzyński 2005). Many respondents had already taken part in other pilgrimages before, mostly to Częstochowa (35%) and to the Marian sanctuary in Swarzewo, near Puck (20%). For 60% of the respondents, it was their first fishermen's pilgrimage; the remaining were usually "experienced" participants of that particular event. The average number of sea pilgrimages in which the pilgrims had participated was slightly under six. The "record-breakers", usually individual old fishermen, were present at all the pilgrimages organised from 1981 to 2004[4].

The well-known people who took part in the Fishermen's Sea Pilgrimage included President Lech Wałęsa, Professor Zbigniew Brzeziński (American President's advisor), Maciej Płażyński (Polish Sejm Marshal), as well as many politicians and celebrities from the world of culture and science. Among regular participants, there are representatives of the Kashubian–Pomeranian Association, as well as local authorities. In 2010, the pilgrimage attracted the participants of the 12th Convention of Kashubians, travelling from different parts of the world (Huras and Necel 2017), even from Canada. In individual years, the pilgrims included visitors from the United States, Japan, and many European countries. Since 1986, the event has also hosted representatives of the church hierarchy—diocese archbishops and bishops. In 2007, priests and monks from all over the world also took part; they arrived in Gdynia to attend the 22nd World Congress of the Apostleship of the Sea.

### 4. Conclusions

In Poland, pilgrimages—religious migrations to a place which is considered sacred—were organised in past centuries and have continued into the 2020s. The participants of the Fishermen's Sea Pilgrimage chose Saints Peter and Paul the Apostles' church in Puck as their destination. On the map of sacred places and pilgrimage trails in Poland, the event described in this article is an example of the growth in pilgrimage in recent decades. It can be assumed that it is an illustration of a progressing trend in contemporary pilgrimaging in Poland, characterised by: (1) alternative modes of making the journey, apart from walking; (2) pilgrimage groups not based exclusively on the parishioners' place of residence; and (3) the performance of both religious and non-religious functions. Although a pilgrimage takes place in a social context, it first of all "expresses the efforts the individual has to make to give meaning and direction to his or her personal existence" (Margry 2008a, p. 33). In the reality of every day, as well as in the reality of pilgrimage in Poland, where pilgrims make their journey mostly by land (by car, coach, on foot), the unconventional form of travelling by sea seems to meet the needs of the modern individual halfway—the individual who wants to be close to nature and needs new stimuli. This can be seen in the large number of people interested in accompanying fishermen during the sea pilgrimage to Puck.

It appears that participation in the Fishermen's Sea Pilgrimage means combining tradition with modern realities. Fishermen living on Puck Bay and Hel Peninsula, known for their deep faith (Borzyszkowski 2001; Obracht-Prondzyński 2001; Mazurek 2021), on the one hand, show their devotion to the Christian faith and Roman Catholic Church, and to their ethnicity and Kashubian character on the other. Similar conclusions were also reached by other authors in their research on this issue (Czerny 2000; Miluszewska 2002; Marciniak 2008). In addition, the survey conducted among pilgrimage participants (Turzyński 2005), as well as the observations continued by one of the co-authors of this study for many years, allow us to identify not only the predominating religious functions of the Fishermen's Sea Pilgrimage but also the non-religious ones, such as the expression of patriotic feelings and community spirit, both in the religious and socio-national dimension. Jackowski (2003) identifies them as typically Polish features of the pilgrimage movement. It shows in the decoration of the boats which the pilgrims sail (banners with slogans, national, papal, and Kashubian flags), as well as in the pilgrims' behaviour and their declarations of religious and non-religious motives for going on the pilgrimage.

Hel Peninsula has been inhabited by old fishing families for years. They devoted their lives to the sea, fish, and their boats (Huras et al. 2022) in the past and many of their

representatives still work in the fishing industry. In today's world, when fish resources are shrinking, the number of fishing boats on the Polish coast is also decreasing. They are scrapped or cut up and used as firewood. Time will tell how many boats stay on the sea and whether or not the fishermen perpetuate the Kashubian fishing traditions, their ethnic distinctiveness, and their ancestors' faith. These factors undergird the tradition and inspire the sea pilgrimage on fishing boats, organised every June across the Small Sea to the church in Puck.

**Author Contributions:** Conceptualization, L.P.; Methodology, L.P.; Resources, T.P.; Writing—original draft, T.P. and L.P.; Visualization, T.P. All authors have read and agreed to the published version of the manuscript.

**Funding:** This research received no external funding.

**Institutional Review Board Statement:** Not applicable.

**Informed Consent Statement:** Informed consent was obtained from all subjects involved in the study.

**Conflicts of Interest:** The authors declare no conflict of interest.

## Notes

[1] Fishing boats from Hel Peninsula, including Edward Muża's vessel, are described in a publication by Huras et al. (2022).

[2] Pomeranka is a large fishing sailboat, popular in Kashubia in the second half of 19th century and in the first half of 20th century (Celarek 2004) (especially popular in Kashubia in the turn of 19th and 20th centuries).

[3] The place of the Kashubian language in Circum Baltic languages is well documented by Motoki Nomachi (2020). The book *Geography of Kashubs* (Mordawski 2018) develops spatial, demographic, social, cultural, and settlement issues. The historical statistics on Kashubs are available in one of the oldest Polish illustrated guide books published in the 1920s (Orłowicz 1924).

[4] Apart from the Sea Pilgrimage to Puck, the Puck Bay fishermen, wishing to express their ties with God and their gratitude for Pope John Paul II's visit, organised a sea pilgrimage to the Holy Mass in Sopot on 5 June 1999. Over 80 fishing boats took part in that pilgrimage; they carried over 1400 pilgrims—fishermen and their families—to the pier in Sopot (Lemak 2003). Most (25) vessels sailed from the port in Jastarnia, and most pilgrims travelled from Hel (472), followed by Jastarnia and Władysławowo. It is estimated that, apart from 1400 Kashubian pilgrims, about 1500–2000 took part in a grand sea people's parade; they were a part of the crews on over 150 yachts, motorboats, and other vessels (Turzyński 2005).

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
