# Peer review of "The Functions of the Fishermen’s Sea Pilgrimage to St Peter and St Paul’s Church Fair in the Town of Puck"

_religions, doi:10.3390/rel13121148_

Round 1

Reviewer 1 Report

The paper titled "The Functions of the Fishermen`s Sea Pilgrimage to St Peter and St Paul`s church fair in the town of Puck '' presents an interesting case study of contemporary pilgrimage practices in Polish context. As the only pilgrimage over sea in Poland this paper has a potential to contribute to broader discussion on sea and pilgrimage interconnections. However, the authors do not reference or discuss with other similar work done on "sea pilgrimages' '. They define the practice as "sea pilgrimage" but do not explain why and what does that actually (re)presents? In order to use their work to contribute to further development of pilgrimage studies in general, I suggest to the author to set their work within other similar work already done and discussions with other authors about the definition and approach to "sea pilgrimage". I suggest that the starting point should be a paper by Katic and McDonald on maritime pilgrimage in Ireland (2020) published in Anthropological Notebooks. The authors should also consider other work done on contemporary pilgrimages in Poland, like the work by Anna Niedźwiedź, for example. I realise that the paper is on "sea pilgrimage" and the focus is different than within usual papers on pilgrimage, but the authors themselves reference other pilgrimage sites and practices in Poland as a relevant framework. In any case, positioning their work and case study within other similar research such as on "maritime pilgrimage" could be very useful for this paper and broader research on "sea pilgrimage". I believe that after taking into consideration similar work, the authors will revise their paper and produce a much stronger contribution.

Author Response

We are pleased to read the positive comments and we really appreciate them. We  considered the Reviewer`s suggestion that „The authors should also consider other work done on contemporary pilgrimages in Poland”. We followed this advice and the following papers have been incorporated to the manuscript:

Mróz F., 2021, Shrines and Pilgrimages in Poland as an Element of the “Geography” of Faith and Piety of the People of God in the Age of Vatican II (c. 1948–1998), Religions 12, no. 7: 525, 1-26.

Niedźwiedź A., 2015, Old and New Paths of Polish Pilgrimages. In: Eade, John and Albera, Dionigi (eds.), International Perspectives on Pilgrimage Studies: Itineraries, Gaps and Obstacles in Pilgrimage Studies. London, New York: Routledge, 69-94.

Futhermore, in the revised manuscript, we followed the Reviewer`s suggestion to explain what "sea pilgrimage" represents. We used a paper by Katic and McDonald, strongly recomennded by the Reviewer as well as a book by Harbison:

Harbison P., 1992, Pilgrimage in Ireland: the monuments and the people, Syracuse University Press, New York.

Katić M, McDonald M., 2020, Experiencing maritime pilgrimage to St Mac Dara Island in Ireland: Pilgrims, hookers, and a local saint, Anthropological Notebooks, 26/2, 1-27.

All changes in a revised manuscript are visible thanks to track changes option.

Reviewer 2 Report

The article concerns an interesting phenomenon of the life of the Church in Poland  and in Europe. It is only such pilgrimage in Europe and one of two in the world. I propose to add some information about what similar pilgrimages looked like in previous centuries. One can mention the harassment against Poles during martial law. It is also worth develop a thread that it is good opportunity to get to know Kashubian folklore. It is also worth saying more about very rich tradition of pilgrimage in Poland, which also justifies the unique pilgrimage of fishermen. There are my suggestions. 

Author Response

We are pleased to read the positive comments and we really appreciate them. We considered the Reviewer`s suggestion to “develop a thread that it is good opportunity to get to know Kashubian folklore”. We used the following books and papers:

Celarek A.,2004, Kaszubskie łodzie, Wydawnictwo BiT, Gdańsk.

Mordawski J., 2018, Geografia Kaszub, Zrzeszenie Kaszubsko-Pomorskie, Gdańsk.

Nomachi, M., 2020, Placing Kashubian in the Circum-Baltic (CB) area, Prace Filologiczne, (74), 315-328.

Orłowicz M., 1924, Ilustrowany przewodnik po Ziemi Kaszubskiej, Książnica Polska, Lviv and Warsaw.

We also considered the Reviewer`s suggestion on a historical context of Polish pilgrimages. It  has been extended by incorporation of the following papers:

Mróz F., 2021, Shrines and Pilgrimages in Poland as an Element of the “Geography” of Faith and Piety of the People of God in the Age of Vatican II (c. 1948–1998), Religions 12, no. 7: 525, 1-26.

Niedźwiedź A., 2015, Old and New Paths of Polish Pilgrimages. In: Eade, John and Albera, Dionigi (eds.), International Perspectives on Pilgrimage Studies: Itineraries, Gaps and Obstacles in Pilgrimage Studies. London, New York: Routledge, 69-94.

However, we decided not to extend a historical context of pilgrimages in Poland too much because it has partly been presented in section 2 “The origins and form of the Fishermen’s Sea Pilgrimage against contemporary trends in pilgrimaging in Poland”.

All changes in a revised manuscript are visible thanks to track changes option.

Round 2

Reviewer 1 Report

Dear authors,

I still think your paper is a very interesting and has potentinal to contribute to research of neglected form of pilgrimages, not just in Polish context, but even broader. My suggestions were in good faith in order to push you to make your paper stronger and more relevant. What you did is just quote the papers and authors I suggested, but not really engaged with their work and give your contribution in the discussion on sea/maritime pilgrimage. That is a shame, but that is your choice. Good luck in your research.

All the best.

Author Response

Dear Reviewer,

We inform that our paper has been proofreaded by a native speaker from Memorial Univerisity of Newfoundland (Canada).

We not only used papers strongly recomennded by the Reviewer but a book by Harbison as well [Harbison P., 1992, Pilgrimage in Ireland: the monuments and the people, Syracuse University Press, New York].

We omitted discussion suggested by a Reviewer. We think it is a topic worth a separate article.
